# BRAINTEASER: Lateral Thinking Puzzles for Large Language Models

**Yifan Jiang[1], Filip Ilievski[1,2], Kaixin Ma[3*], Zhivar Sourati[1]**

[1]Information Sciences Institute, Viterbi School of Engineering, University of Southern California
[2]Department of Computer Science, Faculty of Science, Vrije Universiteit Amsterdam
[3]Tencent AI Lab, Bellevue, WA
{yifjia,ilievski,Souratih}@isi.edu, f.ilievski@vu.nl
kaixinma@global.tencent.com

## Abstract

The success of language models has inspired the NLP community to attend to tasks that require implicit and complex reasoning, relying on human-like commonsense mechanisms. While such vertical thinking tasks have been relatively popular, lateral thinking puzzles have received little attention. To bridge this gap, we devise BRAINTEASER: a multiple-choice Question Answering task designed to test the model's ability to exhibit lateral thinking and defy default commonsense associations. We design a three-step procedure for creating the first lateral thinking benchmark, consisting of data collection, distractor generation, and generation of reconstruction examples, leading to 1,100 puzzles with high-quality annotations. To assess the consistency of lateral reasoning by models, we enrich BRAINTEASER based on a semantic and contextual reconstruction of its questions. Our experiments with state-of-the-art instruction- and commonsense language models reveal a significant gap between human and model performance, which is further widened when consistency across reconstruction formats is considered. We make all of our code and data available to stimulate work on developing and evaluating lateral thinking models.

## 1 Introduction

Human reasoning processes comprise two types of thinking: vertical and lateral (Waks, 1997). Vertical thinking, also known as linear, convergent, or logical thinking, is a sequential analytical process that is based on rationality, logic, and rules, typically associated with the left-brain hemisphere. Vertical thinking, as illustrated in Figure 1 (top), is needed to create a reasoning path from flooding a room to filling it with water for physical reasoning, and from inanimate objects with five fingers to gloves in riddles. Meanwhile, lateral thinking (or "thinking

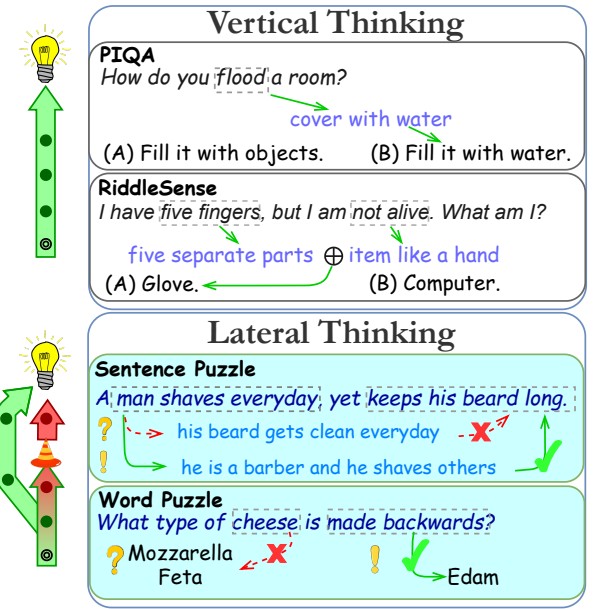

Figure 1: Contrasting existing Vertical Thinking tasks (PIQA (Bisk et al., 2020) and RiddleSense (Lin et al., 2021)) to our novel lateral thinking task called BRAINTEASER. While prior tasks require commonsense to be injected, BRAINTEASER's lateral thinking puzzles require default commonsense thinking to be deprecated.

outside the box") is a divergent and creative process that involves looking at a problem from a new perspective and defying preconceptions, associated with the right-brain hemisphere (De Bono, 1970; Waks, 1997). Lateral thinking is required to solve the puzzle in Figure 1 (bottom), by overwriting the commonsense associations of *man shaves* to *he shaves himself*, and regarding the man as somebody who shaves others all day (e.g., a barber).

The development of natural language processing (NLP) models and their evaluation has achieved much progress in vertical thinking. In particular, large language models (LLMs) (Devlin et al., 2019; Liu et al., 2019; Brown et al., 2020b) have achieved strong performance across a variety of complex reasoning tasks (Talmor et al., 2019; Bisk et al., 2020;

---

* Work done when KM was at Carnegie Mellon University

Sap et al., 2019b), even with the complete absence (zero-shot) (Sanh et al., 2022) or limited provision (few-shot) of training time exemplars (Chung et al., 2022).[1] To perform well on tasks such as reasoning over physical interactions (Bisk et al., 2020) and social implications (Sap et al., 2019b), LLMs exhibit better vertical thinking capabilities, including commonsense association (Wei et al., 2022) and inference ability (Bosselut et al., 2019). While the extent to which these models possess common sense is heavily discussed (Marcus, 2022; Bubeck et al., 2023; Wei et al., 2023), we note that prior work has not considered the lateral thinking ability of LLMs. Creative thinking problems in benchmarks and knowledge bases are often filtered out as noise during preprocessing (Vajjala and Meurers, 2012; Speer et al., 2017; Sap et al., 2019a), and only kept if their resolution can be supported by commonsense associations, as in the case of riddles (Figure 1) (Lin et al., 2021; Gao et al., 2018). As many situations are novel, we expect that lateral thinking puzzles like those in Figure 1-bottom will be hindered by default commonsense associations and cannot be easily solved by further adaptation and scaling of the existing LLM methods.

To bridge this gap, we propose to *study the ability of state-of-the-art LLMs to reason on lateral thinking puzzles*. We formulate lateral thinking puzzles as multiple-choice Question Answering (QA) tasks, making them intuitive to answer by humans and easy to evaluate automatically. Following our task definition, we create a novel BRAINTEASER benchmark with two tasks of different granularity: Sentence Puzzles and Word Puzzles (cf. Figure 1). To construct the dataset, we design a data collection procedure, which crawls relevant puzzles from several publicly available websites, performs semi-automatic filtering of irrelevant question categories (e.g., pun, dad jokes), and ensures high data quality. To ensure fair and informative questions, we construct distractors semi-automatically by manual annotation of the explicit and implicit (commonsense) premises that arise from each puzzle. To address concerns of possible LLM memorization (Carlini et al., 2022) and their lack of consistency (Goldberg, 2023), we enrich BRAINTEASER with two reconstruction strategies: *semantic reconstruction* and *context reconstruction*, which create variants of each puzzle without changing its original way of

defying default commonsense associations. This systematic procedure results in a novel BRAINTEASER benchmark with 1.1K high-quality data points and nearly 100% human evaluation results. Using BRAINTEASER as the benchmark, we conduct comprehensive experiments involving different model structures, model sizes, and prompting strategies. The results reveal a huge gap between human performance and current LLMs, indicating the great need to improve lateral thinking in LLMs.

We summarize our contributions as follows: 1) We introduce **lateral thinking puzzles**, a multiple-choice QA task designed to test the model's ability to exhibit lateral thinking and defy default commonsense associations. 2) We design a three-step procedure for creating **the first lateral thinking benchmark, BRAINTEASER**, consisting of data collection, distractor generation, and generation of reconstruction examples, leading to 1,100 high-quality puzzles. 3) We conduct **comprehensive experiments** with state-of-the-art LLMs. We make all of our code and data available to stimulate work on developing and evaluating lateral thinking models.[2]

## 2 Related work

We review prior work on computational creativity, commonsense reasoning, and model robustness.

**Computational Creativity** Computational creativity work includes a broader set of tasks, some of which have been relatively popular, including pun (Zou and Lu, 2019) and humor (Meaney et al., 2021) detection. A particular class of creative challenges, called *brain teasers* (Draper, 2009; Highhouse et al., 2019), is designed to evaluate a wide range of human intelligence skills, including strategy development, planning, visual-spatial thinking, creativity, and memory (Altun et al., 2016). Most similar to our task, Lin et al. (2021) collects riddles from public websites to challenge current models. While in principle computational creativity puzzles and brain teasers combine vertical and lateral thinking, prior work has focused on the former category. Our BRAINTEASER task complements these works with word- and sentence-level lateral thinking puzzles. BRAINTEASER can serve as a formal platform to evaluate the creative skills of LLMs, which have been partially explored in recent work

---

[1]In this paper, we use the terms *language model* and *large language model* interchangeably.

[2]The code is available at https://github.com/1171-jpg/BrainTeaser

(Franceschelli and Musolesi, 2023; Bubeck et al., 2023; Wang et al., 2023a).

**Commonsense Reasoning** The task of commonsense reasoning has been popular in recent years (Rajani et al., 2019; Ma et al., 2019; Lourie et al., 2021; Maharana and Bansal, 2022), accompanied by the introduction of numerous challenging benchmarks (Talmor et al., 2019; Sap et al., 2019b; Sakaguchi et al., 2019) and availability of large-scale commonsense resources (Speer et al., 2017; Hwang et al., 2021). While each of the existing datasets focuses on different dimensions of commonsense knowledge (Ilievski et al., 2021a), most of them are constructed in the multiple-choice format, due to the ease of evaluation. Some prior works have focused on generative commonsense reasoning (Lin et al., 2020; Boratko et al., 2020). However, due to the vast plausible answer space, the evaluation has been challenging and a large amount of answer annotations have to be collected in order to ensure fairness (Boratko et al., 2020). Curiously, while possession of common sense has been a central goal of AI, its role in our BRAIN-TEASER task is as a distractor. Namely, successful solutions of the lateral thinking puzzles in BRAIN-TEASER require the models to defy commonsense associations and linear inference chains.

**Robustness Studies** As a novel benchmark, BRAINTEASER relates to other works that evaluate the performance of LLMs. Since these models are surpassing human performance on some existing benchmarks (Xu et al., 2022), the NLP community has shifted the focus towards robustness evaluation, i.e., whether the model can retain a similar performance to semantically perturbed or adversarially constructed questions (Abdou et al., 2020; Nie et al., 2020). Some recent works have adopted model adversarial approaches to generate datasets that are challenging for models to solve (Zellers et al., 2019; Sakaguchi et al., 2019), while others combine multiple tasks to evaluate the model's behavioral consistency across semantic, logical, and factual categories (Jang et al., 2022). Besides dataset construction, analysis studies have also shown that models easily learn shortcuts to solve the datasets (Branco et al., 2021; Elazar et al., 2021) and their performance heavily depends on the overlap of tokens between training and test data (Ma et al., 2021b). Different from prior works where associative resources are used to finetune the model to improve robustness, we expect that the lateral thinking puzzles in BRAINTEASER require unique associations and creative reasoning paths. In this way, BRAINTEASER is designed to minimize the impact of confounding factors like memorization in LLMs (Bang et al., 2023; Guo et al., 2023; Goldberg, 2023).

## 3 Construction of BRAINTEASER

In this section, we first provide a definition of lateral thinking puzzles in various granularities. We then present a three-stage pipeline for constructing the multiple-choice puzzles in the BRAINTEASER dataset, consisting of data collection, distractor sampling, and reconstruction sample generation. Finally, we present key data statistics and quality validation results.

### 3.1 Task Definition

While lateral thinking puzzles are often presented to humans in an open-ended fashion, these are difficult to evaluate automatically and are difficult to solve by humans.[3] An additional complication is that there may be multiple independent, yet correct, puzzle explanations. To alleviate these challenges, we pose lateral thinking puzzles as a multiple-choice QA task, a format frequently employed for reasoning tasks. We expect this approach to be both facile for human comprehension and amenable to automated evaluation. In general, each puzzle contains a question $Q$ stating the context, and a lateral explanation $e$ from explanation space $E$ that serves as the correct answer. $Q$ can be decomposed into an atomic premise set $P$, which includes both explicitly stated clauses and implicit clauses derived through default commonsense inferences or associations. For example, in the following puzzle: "*How could a cowboy ride into town on Friday, stay two days, and ride out on Wednesday?*", the set $P$ includes the following premises:

- $p_1$: Cowboy rides into town on Friday.

- $p_2$: Cowboy stays in town for two days.

- $p_3$: Cowboy rides out on Wednesday.

- $p_4$: Wednesday is the third day of the week.

- $p_5$: Sunday is two days after Friday.

---

[3]Our small-scale user study shows that both humans and LLMs are unable to perform this open-ended task well, scoring 2.64 and 2.62 on a 5-point scale, respectively (see Appendix A.5 for details).

The premises $p_1$, $p_2$, and $p_3$ are explicitly provided by the context, and the premises $p_4$ and $p_5$ are implicitly obtained by default commonsense association. The goal of a puzzle is to find an explanation that does not contradict the premise set $P$, $E \cap \neg P = \varnothing$, as the premises are the target to explain and support. With vertical thinking, the question appears impossible to answer because $P$ contains statements that conflict with each other. The premises $p_3$ and $p_4$ are inconsistent with other premises, leading to an obstacle in explaining the puzzle. The default commonsense inference thus becomes a logic stumper (Bar-Hillel et al., 2018), preventing one from creatively exploring additional explanations in $E$.

Lateral thinking leads to a correct solution to this puzzle: "His horse is named Wednesday.". This creative solution defies the commonsense association of Wednesday as a third day of the week ($p_4$). Thus, the key point of a lateral thinking puzzle is that some implicit premises generated through default commonsense association incorrectly create an arbitrary "box" that wrongly excludes the possible solution from the explanation space (Bar-Hillel et al., 2018).

Upon careful exploration, we devise two granularity variants of lateral thinking puzzles following our definition (Figure 1): *sentence-based*, where the puzzle is centered on sentence premises (e.g., *Wednesday is the third day of the week*), and *word-based*, where the answer violates the default meaning of the word and focuses on the letter composition of the target question (e.g., *cheese made backwards → edam*).

## 3.2 Data Collection

We collect over ten thousand lateral thinking puzzles with answers from public websites such as `riddles.com` and `rd.com` using web crawlers. We merge the data from different sources and remove (near-)duplicates based on sentence similarity (Reimers and Gurevych, 2019). We conduct a semi-automatic process that corrects typos by using an automatic library, Auto Correct,[4] followed by human verification to ensure that the puzzles preserve their original meaning. We filter the remaining data manually to preserve QA pairs that fit the definition of the sentence- or word-based lateral thinking puzzles. This process yields 373 unique lateral puzzles, formatted as QA pairs.

---

[4] `github.com/phatpiglet/autocorrect`

Table 1: Example of generated distractors.

| Premise | Answer/Distractor |
|---|---|
| $p_W$: Wednesday is the third day of the week. | **Answer**: His horse is named Wednesday. |
| $p_2$: Cowboy stays in in town for two days. | **Distractor**: While in town, he stays in bed for two days. |
| $p_5$: Sunday is two days past Friday. | **Distractor**: Friday and Saturday are holidays. |

## 3.3 Distractor Sampling

We convert each puzzle and its explanation into a multiple-choice QA format to ensure a straightforward evaluation process. A key challenge in creating fair and informative multiple-choice questions is sampling distractors that are simultaneously incorrect and challenging (Ma et al., 2021a). We propose a systematic approach for distractor sampling that directly benefits from our premise-based definition of lateral thinking puzzles.

For *sentence puzzles*, we list possible premises $P = \{p_1, p_2, p_3, \ldots\}$ from the question context manually as the commonsense associations in the data are obvious and straightforward, especially when the answers are provided, like the example in Section 3.1. We know the correct answer $p'_c$ is an unconventional overwriting of the wrong premise (logic stumper) $p_w$ generated by default commonsense association. We generate the distractors by overwriting other premises in $P - p_w$. This procedure guarantees that the distractors are incorrect because the misleading premise $p_w$ still remains in the premise set and prevents one from reaching the correct explanation. We first use COMET (Hwang et al., 2021) to generate the possible premise overwriting candidates for the question as a head combined with inference relations (e.g., happens after, hindered by, cause). Then we pick the COMET-generated tails that are consistent with the question context as distractors and revise them by manual annotation. Table 1 shows example distractors for our running example puzzle from Section 3.1.

For *word puzzles*, as we focus on the literal meaning rather than semantic meaning, distractors can share similar semantic meaning as the correct answers and still exhibit similar commonsense associations. We pick distractors from the correct answer's synonyms in WordNet (e.g., *mozzarella* for *edam* in Figure 1) and Wikipedia entries that belong to the same category (e.g., both *edam* and *cheddar* belong to the *semi-hard cheese* category).

Since it is generally possible that none of the cre-

Table 2: A sentence-based lateral thinking puzzle and its reconstruction variations. We present an analogous word-level puzzle in the Appendix A.3.

| Adv Strategy | Question | Answers |
|---|---|---|
| - | How could a cowboy ride into town on Friday, stay two days, and ride out on Wednesday? | **His horse is named Wednesday.** While in town, he stays in bed for two days. Friday and Saturday are holidays. None of the above. |
| Semantic Reconstruction | How could a cowboy come into town on Friday, stay two days, and then ride away on Wednesday? | **His horse is named Wednesday.** While in town, he stays in bed for two days. Friday and Saturday are holidays. None of the above. |
| Context Reconstruction | How can a pilot take off in Los Angeles on Tuesday, fly for 48 hours, and land in Tokyo on Tuesday? | **The pilot's airplane is named Tuesday.** He flies straight for 24h and flies quickly for hours left. There was a one-week long holiday. None of the above. |

ative solutions will be sensible for some of the questions, we also include the option *None of the above* in all questions' candidates set. This answer candidate simulates the situation where humans cannot overwrite their commonsense inference and give up on explaining the lateral thinking puzzle. To create puzzles where lateral thinking fails (i.e., with answer *None of the above*), we replace the correct answer with a distractor in 6% of the questions. After this procedure, each question in BRAINTEASER has four answer candidates.

### 3.4 Generating Reconstruction Examples

Since the latest LLMs are pretrained on massive web snapshots, it is possible that the data sources for BRAINTEASER are also included in their training data. Consequently, it is possible for LLMs to memorize the correct answer without performing any reasoning. To ensure that our task evaluates lateral thinking ability rather than memorization, we construct reconstruction versions of the original data in two parallel ways (Table 2): (1) *Semantic Reconstruction* rephrases the original question without changing its answer, distractor, and any premises in $P$. To do so, we use an open-source rephrasing tool,[5] after which human annotators refine and validate that all premises remain the same. (2) *Context Reconstruction* keeps the misleading commonsense premise intact and changes both the question and the answer to a new situational context. For this purpose, we prompt GPT-4 for initial reconstructions, which are then manually refined by human annotators. The new distractors are generated following the same process as in Section 3.3. The premise set and the corresponding distractors also get translated to the new context. Intuitively, a

Table 3: Key statistics of the BRAINTEASER dataset. Choices combine the correct answer with all the distractors. Standard deviation is computed without the *None of the above* choice, as its token length is fixed and not related to the question context.

|  | Sentence | Word |
|---|---|---|
| # Puzzles | 627 | 492 |
| Average Question Tokens | 34.88 | 10.65 |
| % Long Question (>30 tokens) | 48.32% | 2.23% |
| Average Answer Tokens | 9.11 | 3.0 |
| Std of Choice Tokens | 2.36 | 0.52 |

model that learns to reason should be able to solve these two reconstruction variants of the questions easily, whereas the model that memorizes the answer would stumble.

### 3.5 Data Analysis and Validation

**Key Statistics** BRAINTEASER includes 1,119 data samples including its reconstruction variants. Table 3 reports key statistics of each subtask of BRAINTEASER. The questions in the *Sentence Puzzle* category are much longer because they are in a narrative story format rather than simple short questions, like those in the *Word Puzzle* category. The difference between the standard deviation in the number of choice tokens between *Sentence Puzzle* and *Word Puzzle* can be ascribed to the different strategies for generating distractors, i.e., overwriting various premises with new statements versus generating similar words from the synonym set.

We use ChatGPT prompting to extract the context topic from each question and to analyze the major topics in each subtask. The topic distribution shows that both subtasks involve a large range of (more than 80) areas. *Sentence Puzzle* is denominated by math, physics, and nature while *Word Puzzle* is denominated particularly by the language

---

[5] https://quillbot.com/

topic. For both tasks, there is a long tail of less common topics. The details of topic extraction and its obtained statistics are given in the Appendix A.1. The data statistics and the topic analysis suggest that, despite its limited size, BRAINTEASER can function as a comprehensive benchmark for assessing model performance across diverse topics and varying lengths of context.

**Human Validation** To ensure the quality of our dataset, we invited three expert annotators to verify the validity of the QA pairs and their reconstruction variants. We sampled 102 examples from BRAINTEASER randomly and asked the annotators the following two questions: 1) Does the original puzzle and correct answer make sense? 2) Are the reconstruction variants still consistent with the original questions in terms of the required reasoning process? On average, the human annotators rated 99% of the original question-answering pairs as valid. 100% of the semantic reconstructions and 97% context reconstructions were marked as consistent with the original question-answer pair. The overall Fleiss (1971) kappa inter-annotator agreement is 0.948, which is an almost perfect score.

## 4 Experimental Setup

We describe the models selected for our experiments and the metrics used to evaluate the reasoning accuracy and consistency of these models.

### 4.1 Model selection

**Instruction-Based Models** We evaluate the instruction-finetuned LLMs in zero/few-shot setting: 1) ChatGPT, a publicly available state-of-the-art LLM from the GPT (Brown et al., 2020a) series. 2) T0 (Sanh et al., 2022), a LLM trained with multitasking instruction tuning that has strong zero-shot generalization ability. 3) FlanT5 (Chung et al., 2022), an enhanced version of T5 (Raffel et al., 2020) which is instruction-finetuned (Wei et al., 2021) in both zero-shot and few-shot setting. For a fair comparison with humans, while running zero-shot prompting on ChatGPT, we add a description indicating that the question is a brain teaser puzzle that needs creative thinking to solve. For the rest of the models, we use the same instruction templates as found in their training sets (for full details, please refer to Appendix A.2).

**Commonsense Models** To understand the effect of commonsense knowledge on our task, we evaluate the following models that are enhanced with

common sense: 1) RoBERTa-L (CSKG) (Ma et al., 2021a), a model finetuned on the synthetic QA pairs generated from a diverse set of commonsense knowledge graphs (CSKG) (Ilievski et al., 2021b). 2) CAR (Wang et al., 2023b), a model finetuned in a similar pipeline as Ma et al. (2021a) but with enhanced negative sampling strategy and reportedly superior performance. For reference, we also include the vanilla RoBERTa model (Liu et al., 2019) to understand the impact of commonsense knowledge. We evaluate all of the models in a zero-shot fashion, following the scoring method defined in (Ma et al., 2021a). We select RoBERTa because of its widespread usage of the commonsense task and impressive zero-shot performance. RoBERTa-L (CSKG) achieve SOTA zero-shot result on multiple commonsense tasks, while CAR even outperforms ChatGPT on commonsense tasks.

**Human Evaluation** To assess the upper bound performance on BRAINTEASER, we randomly sample 102 questions from it and invite three experts annotator to solve the test. On average, it takes one hour for an annotator to complete the task.

### 4.2 Evaluation Metrics

As accuracy is a fair evaluation metric for the MCQA format and it has been adopted by many popular commonsense reasoning tasks (Mihaylov et al., 2018; Talmor et al., 2019; Bisk et al., 2020), we evaluate model performance using two accuracy metrics: **Instance-based Accuracy** considers each (original or reconstruction) question separately. We report instance-based accuracy on the original puzzles, and their semantic and context reconstructions. **Group-based Accuracy** considers each original puzzle and its variants as a group. The model will score 1 only when it successfully solves all three puzzles in the group, otherwise, its score is 0.

## 5 Results

Our experiments target five questions: 1) Can LLMs reason on lateral thinking puzzles similar to humans? 2) How do LLMs perform on reconstruction variants? 3) Are model predictions consistent across partitions? 4) Does tuning on commonsense knowledge help to answer BRAINTEASER puzzles better? 5) Can LLMs do better in the few-shot setting with more demonstrations?

**Overall Performance** The main results are shown in Table 4. For both word and sentence BRAINTEASER puzzles, the performance of the

Table 4: Main zero-shot results over two BRAINTEASER subtasks across all models in all metrics: Ori = Original, Sem = Semantic, Con = Context. The best performance among all models is in bold, and the best performance in commonsense augmented models is underlined. The human evaluation (*) is computed over 102 randomly sampled data. The random base is average over three different seeds.

| Category | Model | Instance-based | | | Group-based | | Overall |
|---|---|---|---|---|---|---|---|
| | | Original | Semantic | Context | Ori & Sem | Ori & Sem & Con | |
| *Sentence Puzzle* | | | | | | | |
| **Random** | - | 25.52 | 24.88 | 22.81 | 5.58 | 1.44 | 24.40 |
| **Instruction** | FlanT5(780M) | 18.66 | 16.27 | 22.01 | 10.53 | 4.31 | 18.98 |
| | FlanT5(3B) | 26.79 | 25.36 | 35.41 | 20.10 | 12.92 | 29.19 |
| | FlanT5(11B) | 33.49 | 31.58 | 36.84 | 22.01 | 11.00 | 33.97 |
| | T0(11B) | 22.01 | 22.01 | 29.67 | 16.27 | 11.00 | 24.56 |
| | T0P(11B) | 23.92 | 22.49 | 34.93 | 17.70 | 11.96 | 27.11 |
| | T0PP(11B) | 26.32 | 27.27 | 37.80 | 19.14 | 11.96 | 30.46 |
| | ChatGPT | **60.77** | **59.33** | **67.94** | **50.72** | **39.71** | **62.68** |
| **Commonsense** | RoBERTa-L | 43.54 | 40.19 | 46.41 | 33.01 | 20.10 | 43.38 |
| | RoBERTa-L(CSKG) | 35.41 | 36.84 | 44.98 | 28.71 | 18.18 | 39.07 |
| | CAR | 10.53 | 10.53 | 11.48 | 5.74 | 2.39 | 10.85 |
| **Human*** | - | 90.74 | 90.74 | 94.44 | 90.74 | 88.89 | 91.98 |
| *Word Puzzle* | | | | | | | |
| **Random** | - | 26.02 | 27.85 | 22.51 | 7.32 | 1.83 | 25.34 |
| **Instruction** | FlanT5(780M) | 22.56 | 17.68 | 28.66 | 9.15 | 3.66 | 22.97 |
| | FlanT5(3B) | 37.80 | 29.88 | 42.68 | 23.17 | 12.80 | 36.79 |
| | FlanT5(11B) | 42.68 | 32.93 | 43.90 | 28.66 | 20.12 | 39.84 |
| | T0(11B) | 17.07 | 14.02 | 23.17 | 9.76 | 6.10 | 18.09 |
| | T0P(11B) | 28.66 | 26.22 | 34.15 | 19.51 | 12.80 | 29.67 |
| | T0PP(11B) | 33.54 | 31.10 | 39.63 | 20.12 | 10.98 | 34.76 |
| | ChatGPT | **56.10** | **52.44** | **51.83** | **43.90** | **29.27** | **53.46** |
| **Commonsense** | RoBERTa-L | 19.51 | 19.51 | 23.17 | 14.63 | 6.10 | 20.73 |
| | RoBERTa-L(CSKG) | 18.90 | 16.46 | 30.49 | 12.80 | 6.10 | 21.95 |
| | CAR | 38.41 | 31.10 | 20.12 | 26.22 | 6.10 | 29.88 |
| **Human*** | - | 91.67 | 91.67 | 91.67 | 91.67 | 89.58 | 91.67 |

strongest model, ChatGPT (53 and 63%) is halfway between random (25%) and human performance (92%). In general, neither type of model is able to perform consistently well across the two subtasks: instruction-based models perform better on word puzzles, whereas commonsense models perform slightly better on sentence puzzles. The performance of the models is often close to random, with around a third of the models performing equal or worse than random guessing. As it can be expected, we see that scaling up instruction-finetuned models leads to improved performance on both subtasks. Yet, the large gap between human and model performance clearly shows that even the most powerful LLMs are unable to exhibit lateral thinking in multiple-choice puzzles and confirms the challenging nature of our BRAINTEASER dataset.

**Original vs Reconstruction Partitions** In most cases, all models and humans perform the best on the context reconstruction partition. We hypothesize that this is because original lateral thinking puzzles are designed to mislead humans to a wrong choice based on commonsense associations, often involving rare words and unconventional sentence structures. Meanwhile, we note that our contextual reconstruction mechanism yields puzzles that are more familiar or easier to solve than the original puzzle, possibly because some of the commonsense associations are relatively weaker. An exception to this trend is ChatGPT's performance on *word puzzles*, where ChatGPT performs the best on the original examples. We believe that this is due to a combination of two factors. First, the word puzzle reconstructions only have a limited impact on the vocabulary domain and sentence structure, because of the much shorter questions. Second, ChatGPT may have memorized some of the word puzzles, e.g., given the question *"How do you spell COW in thirteen letters?"*, its answer begins with *"The question seems to be a brain teaser ..."* We provide representative examples of the prevalent lateral thinking errors of memorization and commonsense associations in Table 5.

**Consistency of Model Predictions** We further compare the performance on instance- and group-based metrics to understand whether the models can solve lateral thinking puzzles by following a consistent reasoning path. A model understand-

Table 5: Error analysis on memorization and commonsense association.

| Question | Answer | LLM choice |
|---|---|---|
| *Memorization* | | |
| The man calls his dog on the other side of the river, and the dog crosses the river without getting wet and using ant tools. | The river was frozen. | The river was frozen. |
| The man had to cross the rivers. He can't swim or use any tools like the bridge. How does the man succeed in the end? | The river was frozen. | He jumped a half-mile far to across the river. |
| *Commonsense Association* | | |
| What animal has no wings, but yet will fly? | A caterpillar. | An eagle. |
| There is no light on the road and the car's headlight is broken. How can the driver see the black dog? | It was daytime. | The driver is good at listening . |
| How can Jenny read in a totally no light house at night? | The book is in Braille. | It was daytime. |

ing rather than memorizing the reasoning path of the original brain teaser should be able to answer its adversarial reconstructions with ease. Notably, human performance only has a minimal drop on group-based metrics whereas all models suffer significant drops. Further analysis (see Appendix A.6) reveals that ChatGPT and RoBERTa-L fail to answer many (45 and 61%, respectively) of the original or semantically changed puzzles when contextually translated puzzles are solved correctly. These observations suggest that the ability of the models to perform consistent lateral thinking is far from human ability.

**Impact of Commonsense Knowledge**   We observe that commonsense knowledge has a salient negative impact on the model's performance on *sentence puzzles*. The best-performing model in the commonsense category is the vanilla RoBERTa model, whose adaptation with commonsense knowledge leads to a significant drop in results, especially with the CAR method. This trend confirms our initial hypothesis that learning commonsense associations is generally detrimental to complex lateral thinking tasks. Commonsense knowledge has a limited positive impact on the *word-puzzle* task, possibly because much of the commonsense associations learned by these models hold between words, including synonyms. Finally, given the apparent similarity of riddles and lateral thinking puzzles, we finetuned a RoBERTa model on the Riddle-Sense dataset and evaluated it on our task. Again, we observe that the model struggles on solving the puzzles despite gaining better results compared to the vanilla RoBERTa model (see Appendix A.7).

**Impact of Few-Shot Demonstrations**   As LLMs are good few-shot learners (Brown et al., 2020b), we are interested to see if in-context learning can help them better solve our task. We experiment with our two most powerful models: ChatGPT

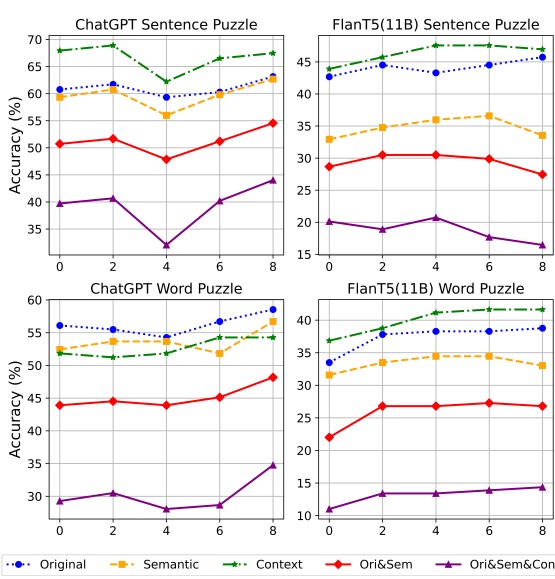

Figure 2: Few-shot prompting performance of ChatGPT and FlanT5(11B).

and FlanT5 (11B). We randomly pick 8 puzzles (4 from each subtask) and create new context reconstructions as demonstrations. We experiment with few-shot prompting with 2, 4, 6, and 8 of these demonstrations, balanced between the two subtasks. The few-shot results are shown in Figure 2, and we present the full results in Appendix A.4. The number of few-shot demonstrations has no clear impact on *sentence puzzles*, which confirms that lateral thinking puzzles are unique and the models can hardly learn generalizable patterns from in-context examples. Providing more few-shot demonstrations has a marginal positive impact for *word puzzles*. Given this task's focus on the letter composition of each word, the in-context examples may be used to teach the model to pay attention to the surface form of the answer candidates. It's worth noting that, although few-shot examples might exhibit superficial resemblances, their contribution to model generalization for sentence puzzles remains

minimal, given the abstract nature of reasoning pattern in this subtask.

**Qualitative Error Analysis** We analyze two prevalent lateral thinking errors in the ChatGPT and FlanT5 (11b) LLMs: memorization and commonsense associations, both of which become more apparent with scaling up (Carlini et al., 2022). We show examples in Table 5.

**Memorization** We find that memorization happens in both subtasks. Given the *sentence puzzle* "*The man calls his dog on the other side of the river, crosses the river without getting wet and using ant tools.*" the LLMs picked the correct answer "*The river was frozen.*" for both the original and its semantic reconstruction. However, when the question in a new context becomes "*The man had to cross the rivers. He can't swim or use any tools. like the bridge. How does the man succeed in the end?*", all LLMs failed to answer. Memorization is more frequent in *word puzzles*. A semantic reconstruction will cause confusion in the model, as is also apparent from the gap between original accuracy and the ori&sem accuracy in Table 4.

**Commonsense association** Similarly, we also find that commonsense association often confuses LLMs. For example, for "*What animal has no wings, but yet will fly?*", the models associate the words "*wings*" and "*fly*" with birds and pick the wrong answer "*An eagle.*" despite the contradiction between "*eagle*" and "*no wings*". Meanwhile, the correct lateral thinking answer "*A caterpillar.*" is not picked by the models. Interestingly, commonsense associations that mislead models in some examples can be the needed hint in others. For example, in one puzzle, "*There is no light on the road and the car's headlight is broken. How can the driver see the black dog?*", the answer "*It was daytime.*" is hindered by the commonsense association between mentioning *no light* and *night*. However, in another example, "*How can Jenny read in a totally no light house at night?*", the same commonsense association leads the model to the correct answer: "*The book is in Braille.*". In the second example, the answer is misled by another commonsense association related to reading.

## 6 Conclusions and Outlook

We defined the task of lateral thinking for LLMs, formulated as a multiple-choice QA with a sentence- and word-level puzzles. We developed BRAINTEASER, a 1.1K lateral thinking benchmark that combines original puzzles and their reconstruction variants. Our experiments showed that ChatGPT's performance on this task is halfway between random and humans, whereas other models often perform close to random. While scaling up model size improved performance, enriching with common sense or providing few-shot demonstrations yielded limited benefits. Meanwhile, all models tend to solve the variants of the same puzzle inconsistently. Our error analysis showed that the models' lateral thinking is often hindered by memorization and misleading commonsense associations. In the future, we intend to develop lateral thinking models, create additional lateral thinking evaluation tasks (e.g., relating to alteration (De Bono, 1970)), and investigate flexible ways to combine lateral and vertical thinking.

## Limitations

While our work focuses on both *Sentence puzzles* and *Word puzzles*, we intend to develop a comprehensive lateral thinking benchmark according to de Bono's four skills: awareness, random stimulation, alternatives, and alteration (De Bono, 1970). Moreover, while our paper tries to provide a clear distinction between lateral and vertical thinking, it remains an open question to which extent other brain teaser categories, e.g. puns and visual puzzles, require lateral or vertical thinking. As these tasks are not the focus of our present paper, we leave it to future work to comprehensively evaluate models' ability to think out of the box on such tasks and to characterize the complementary and opposing aspects of vertical and lateral thinking.

Also, we opt for constructing the dataset in a multiple-choice QA format to reduce the burden of the evaluation process. However, this inevitably reduces the difficulty of the task and permits the situation where the models solve the questions correctly for the wrong reasons. Future work should also look into better evaluation metrics that are suitable for creative and open-ended generations such that our task can also be adapted to an open-ended setting. Finally, while our current puzzles are provided in a static manner, future work should also investigate an interactive (multi-step) setup, where the model (or human) may ask clarification questions or receive contextual hints.

## Ethical Considerations

As our lateral thinking puzzles are "folk knowledge" and are published on a range set of websites, it is hard to check their original licenses comprehensively. Yet, the website owners declare permission to print and download material for **non-commercial use** without modification on the material's copyright. Therefore, we provide the corresponding copyright statements and website URLs for each original lateral thinking puzzle and its reconstruction version. In addition, we will create a form to ask future dataset users to sign a document claiming that the only aim of the data usage is research before providing them with the data. We note that, despite our best efforts, the task data may still contain bias in terms of gender or politics. We will indicate that future research should use the task data with caution.

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

# A  Appendix

## A.1  Puzzle topics

We use few-shot prompting in ChatGPT to extract the context topic for each question. Table 6 shows the top 10 topics for each subtask, for which the prompting template is as follows:
" Can you provide context environment in the following brain teasers? Here are several examples: {examples}"

Table 6: Top-10 topics extracted for both subtasks.

| Sentence Puzzle | | Word Puzzle | |
| --- | --- | --- | --- |
| Topic | Frequency | Topic | Frequency |
| Mathematics | 45 | Language | 79 |
| Physics | 41 | Food | 27 |
| Nature | 37 | Mathematics | 26 |
| Transportation | 32 | Animals | 24 |
| Animals | 25 | Science | 22 |
| Sports | 24 | Time | 18 |
| Family | 23 | Geography | 16 |
| Time | 19 | Nature | 13 |
| Education | 16 | Entertainment | 12 |
| Law | 16 | Finance | 11 |
| Others | 339 | Others | 244 |

## A.2  Prompting templates

**ChatGPT** We use the following instruction to prompt ChatGPT:
*"Please pick the best choice for the brain teaser. Each brain teaser has only one possible solution, including the choice none of the above, answer should only provide the choice:"*

**FlanT5** We use the instruction template for the AI2 Reasoning Challenge (ARC) (Clark et al., 2018):
*"Question: {Question}*

*What is the correct answer to the question from the following choices?*
*(A) {choice}*
*(B) {choice}*
*(C) {choice}*
*(D) {choice}"*

**T0PP** We use the instruction template for the CommensenseQA task (Talmor et al., 2019):
*"{Question}*
*Choose the most suitable option to answer the above question.*
*Options:*
*A. {choice}*
*B. {choice}*
*C. {choice}*

*D. {choice}"*

## A.3  Word puzzle example

Table 7 presents a word puzzle with its reconstruction examples.

## A.4  Few-shot prompting result

Table 8 shows the few-shot result of ChatGPT and FlanT5 (11B) on the two BRAINTEASER subtasks.

## A.5  Annotation Details

**Human evaluation** We give the following instruction to human evaluation participants:

*"Hi, welcome to the brain teaser test. Each brain teaser has only one possible solution (none of the above is possible!). Please select the choice in the answer column. Try to Think out of Box :)"*

**Human validation** We give the following instruction:

*"Congratulations on passing the brain teaser test. You should notice that some brain teasers are similar to each other :)! Actually, the brain teasers can be divided in groups like the following: In each brain teaser group, we have an original question, semantic reconstruction questions, and context reconstruction questions. Semantic reconstruction question rephrases the original question without changing the correct answer and the distractors. Context reconstruction question keeps the original reasoning path but changes both the question and the answer to describe a new situational context.*

*Please help with the following three tasks: 1)Whether the original question and its answer make sense. 2)Whether the semantic reconstruction question rephrases the original question. 3)Whether the context reconstruction question keeps the original reasoning path."*

**Open-ended Human Performance** We give the following instruction:

*"Please write down the answer of each brain teaser. Anything that makes sense is welcome!! Also, no answer is acceptable!"*

We let both humans and ChatGPT write down the most possible answer to 30 context reconstruction questions based on their understanding. Three experts score the answers on a scale of 5, based on the following rubrics:

- **score 0**: Fail to answer.

- **score 1**: Try to answer the question, but the answer doesn't make sense.

- **score 2**: The answer is wrong but related to the golden label.

- **score 3**: The answer is wrong, but the reasoning strategy is similar to the golden answer and may lack some keywords.

- **score 4**: The answer is wrong but lacks minor information. Or the answer makes sense but is not the same as the golden answer.

- **score 5**: The answer is correct.

Both humans and LLMs cannot perform this task well, scoring 2.64 and 2.62 on a 5-point scale. Humans give up more often (18%) rather than generating meaningless text like ChatGPT, making the comparison harder if the task is in an open-end format.

## A.6  Evidence of stronger distractors in the original puzzle

The barber example in Figure 1, "*shaves everyday*" and "*keep his beard long*" triggers a commonsense association that the man shaves himself every day. The contextually reconstructed puzzle of the barber example is "*How can a man go to football team every day but doesn't play football at all?*". This new question still aims to guide the model to think in the default commonsense way that "*He is a football player.*" but the correct answer "*He is a coach.*" is also highly probable, resulting in an inherent decrease in difficulty.

## A.7  Fine-tuned on Riddle Sense

We finetuned RoBERTa-L on RiddleSense (Lin et al., 2021) to analyze whether being aware of linguistic creativity can enhance the model's performance on BRAINTEASER. We train RoBERTa-L (RS) on the training data of RiddleSense in 3 epochs with a learning rate at $1e^{-6}$, batch size at 4. RoBERTa-L (RS) reaches 59.95 on the validation set, which is on par with the original paper (60.72). We then adapt Roberta-L (RS) to do the zero-shot evaluation on BRAINTEASER. The results are shown in Table 9.

Even though Roberta-L (RS) already gains insight into creative thinking, it is still struggling on BRAINTEASER. The better results show that enhancing creative thinking during the training may

Table 7: Overview of a word puzzle example and its reconstruction versions.

| Adv Strategy | Question | Answers |
|---|---|---|
| - | What part of London is in France? | **The letter N.** 
 The letter O. 
 The letter L. 
 None of the above. |
| Semantic Re-construction | Which area of London is inside France? | **The letter N.** 
 The letter O. 
 The letter L. 
 None of the above. |
| Context Re-construction | What part of Korea is in China? | **The letter A.** 
 The letter E. 
 The letter R. 
 None of the above. |

Table 8: Main few-shot results of ChatGPT and FlanT5(11B) on two BRAINTEASER subtasks. Ori = Original, Sem = Semantic, Con = Context. The best performance among all models is in bold.

| | Instance-based | | | Group-based | | Overall |
|---|---|---|---|---|---|---|
| **Model** | **Original** | **Semantic** | **Context** | **Ori & Sem** | **Ori & Sem & Con** | |
| *Sentence puzzle* | | | | | | |
| ChatGPT(zero-shot) | 60.77 | 59.33 | 67.94 | 50.72 | 39.71 | 62.68 |
| ChatGPT(two-shot) | 61.72 | 60.77 | **68.90** | 51.67 | 40.67 | 63.80 |
| ChatGPT(four-shot) | 59.33 | 55.98 | 62.20 | 47.85 | 32.06 | 59.17 |
| ChatGPT(six-shot) | 60.29 | 59.81 | 66.51 | 51.20 | 40.19 | 62.20 |
| ChatGPT(eight-shot) | **63.16** | **62.68** | 67.46 | **54.55** | **44.02** | **64.43** |
| FlanT5(zero-shot) | 33.49 | 31.58 | 36.84 | 22.01 | 11.00 | 33.97 |
| FlanT5(two-shot) | 37.80 | 33.49 | 38.76 | 26.79 | 13.40 | 36.68 |
| FlanT5(four-shot) | 38.28 | 34.45 | 41.15 | 26.79 | 13.40 | 37.96 |
| FlanT5(six-shot) | 38.28 | 34.45 | 41.63 | 27.27 | 13.88 | 38.12 |
| FlanT5(eight-shot) | 38.76 | 33.01 | 41.63 | 26.79 | 14.35 | 37.80 |
| *Word puzzle* | | | | | | |
| ChatGPT(zero-shot) | 56.10 | 52.44 | 51.83 | 43.90 | 29.27 | 53.46 |
| ChatGPT(two-shot) | 55.49 | 53.66 | 51.22 | 44.51 | 30.49 | 53.46 |
| ChatGPT(four-shot) | 54.27 | 53.66 | 51.83 | 43.90 | 28.05 | 53.25 |
| ChatGPT(six-shot) | 56.71 | 51.83 | **54.27** | 45.12 | 28.66 | 54.27 |
| ChatGPT(eight-shot) | **58.54** | **56.71** | **54.27** | **48.17** | **34.76** | **56.50** |
| FlanT5(zero-shot) | 42.68 | 32.93 | 43.90 | 28.66 | 20.12 | 39.84 |
| FlanT5(two-shot) | 44.51 | 34.76 | 45.73 | 30.49 | 18.90 | 41.67 |
| FlanT5(four-shot) | 43.29 | 35.98 | 47.56 | 30.49 | 20.73 | 42.28 |
| FlanT5(six-shot) | 44.51 | 36.59 | 47.56 | 29.88 | 17.68 | 42.89 |
| FlanT5(eight-shot) | 45.73 | 33.54 | 46.95 | 27.44 | 16.46 | 42.07 |

be a possible solution to defying commonsense. Yet, we note that the performance of this model also declines on the group-based metrics. Moreover, we point out the possible data distribution overlap between BRAINTEASER and RiddleSense, as RiddleSense was collected publicly from similar online websites and contains much more samples (5.7k) than BRAINTEASER.

## A.8 Human Annotator Information

Our human annotators major in computer science come from East Asia, Europe and the Middle East. All annotators all fluent in English.

Table 9: RoBERTa-L (RS) zero-shot results over two BRAINTEASER subtasks.

| Model | Instance-based | | | Group-based | | Overall |
|---|---|---|---|---|---|---|
| | Original | Semantic | Context | Ori & Sem | Ori & Sem & Con | |
| *Sentence Puzzle* | | | | | | |
| RoBERTa-L(RS) | 42.11 | 45.93 | 54.55 | 37.32 | 27.75 | 47.53 |
| *Word Puzzle* | | | | | | |
| RoBERTa-L(RS) | 23.78 | 26.22 | 31.10 | 20.73 | 9.76 | 27.03 |