# OpenReview forum: "BRAINTEASER: Lateral Thinking Puzzles for Large Language Models"
_EMNLP/2023/Conference — EMNLP 2023 Main_

### Official Review · Reviewer_KrFG · 2023-07-25

**Soundness:** 4

**Excitement:**

4: Strong: This paper deepens the understanding of some phenomenon or lowers the barriers to an existing research direction.

**Paper Topic And Main Contributions:**

The paper proposes a new multiple-choice question answering task, which is intended to test lateral rather than vertical reasoning. They use instruction and common-sense language models to learn to perform the task of lateral thinking.

**Questions For The Authors:**

Line 370: The intuition seems plausible, however it is not completely clear that the model is not relying on a memorized instance to solve a paraphrased task. It would be interesting to see whether there is any work that tests this approach on models where all the (pre)training data is known so that we can see whether this method definitely counteracts the problem of memorization. I am also not convinced that the term ‘adversarial’ applies to such paraphrases, as they do not have different labels from the original items.

Line 530: It is not clear how the authors test the whether the models follow a reasoning path, could you elaborate on this point?

**Reasons To Accept:**

- Interesting idea of using common-sense reasoning for distractor generation, since it is usually one of the goals of LLMs, as the authors note.

- A novel way of generating distractors by using COMET for sentence puzzles


**Reasons To Reject:**

- The evaluation metrics selected are not very reliable. Accuracy is well known to not be a good indication of good performance, especially if there is imbalance in the dataset. It would be more informative to know how the models score on other metrics such as F1.


**Reproducibility:**

4: Could mostly reproduce the results, but there may be some variation because of sample variance or minor variations in their interpretation of the protocol or method.

**Reviewer Confidence:**

2: Willing to defend my evaluation, but it is fairly likely that I missed some details, didn't understand some central points, or can't be sure about the novelty of the work.

---

> ### Author Rebuttal · Authors · 2023-08-29
>
> We appreciate your recognition of the value of our research topic! We really appreciate your suggestions, and here we list the main response in relation to your comments:
>
> ### Evaluation Metrics:
> Accuracy is a fair evaluation metric for the MCQA format and it has been adopted by many popular commonsense reasoning tasks [1,2,3]. As the original question, semantic reconstruction and context reconstruction have different levels of difficulty, we propose Instance-based and Group-based accuracy to test whether the model performs consistently in cascade. F1 score is not adaptable to MCQA tasks, yet, we would be happy to include F1 scores in the final paper if the reviewer can provide more detail on their idea.
> ### Model Memorization (Q.A):
> While it is impossible to guarantee that the model is not solving a brain teaser by memorization, our reconstruction methods address memorization by testing whether the model’s performance is consistent over the group of reconstructed questions. A model understanding rather than memorizing the reasoning path of the original brain teaser should be able to answer its adversarial reconstructions with ease. The human performance shows the method is reasonable (Line 530-532), while language model performance drops significantly on the group accuracy metrics.
> ### Use of the Term ‘Adversarial’ (Q.A):
> While we understand your consideration of the term 'adversarial' in the context of our paraphrases, we state clearly that we use the term ‘adversarial’ to refer to reconstructions of the original question (Line 354-365), we understand the reviewer's point. We will gladly replace this term with ‘reconstruction’, or another term that may be suggested by the reviewer.
> ### Testing the Model's Reasoning Path (Q.B):
> We test whether models follow a consistent reasoning path by comparing the model’s performance on instance- and group-based metrics (Line 526-530). The instance-based metric indicates the model’s understanding of single questions and the group-based metric test whether the model understand the whole group of original question and its adversarial examples. As our two adversarial strategies won’t change the original questions’ reasoning path (Line 098-105), a model should also solve two adversarial variants if it understands the original reasoning path (Line 368-372). Like humans, a consistent performance between instance-based metrics and group-based metrics can indicate the model follows a reasoning path (Line 530-532).
>
> [1] [CommonsenseQA: A Question Answering Challenge Targeting Commonsense Knowledge](https://aclanthology.org/N19-1421) (Talmor et al., NAACL 2019)
>
> [2] [Think you have Solved Question Answering? Try ARC, the AI2 Reasoning Challenge](https://arxiv.org/abs/1803.05457) (Clark et al., arxiv 2018)
>
> [3] [Can a Suit of Armor Conduct Electricity? A New Dataset for Open Book Question Answering](https://aclanthology.org/D18-1260) (Mihaylov et al., EMNLP 2018)

---

### Official Review · Reviewer_c7pC · 2023-08-05

**Soundness:** 3

**Excitement:**

4: Strong: This paper deepens the understanding of some phenomenon or lowers the barriers to an existing research direction.

**Paper Topic And Main Contributions:**

This paper proposes a multiple-choice question answering dataset focusing on lateral thinking (thinking outside the box) puzzles rather than vertical thinking. They design a series of procedures to collect data and pick distractors, and human efforts are involved in filtering data and picking/generating distractors for good quality.

As the puzzles are possibly in training data of models, they construct two adversarial versions, i.e., rephrasing the question, or reconstructing the puzzle to some new situational context. A subset of the dataset, including the adversarial versions, is annotated by humans which shows the dataset is of good quality.

They also test both instruction-based and commonsense models on the constructed dataset and evaluate by instance-based and group-based accuracy. They find that this dataset is very challenging for current models, models are lacking consistency across variants of the same puzzle, and commonsense knowledge and few-shot in-context learning have limited benefits.






**Questions For The Authors:**

A. Have you considered classifying the dataset into more nuanced question types? I'm curious to know whether the model could be benefited from few-shot examples if the given examples are more similar to test examples.

B. I'm curious about the exploration regarding common sense knowledge. Rather than "misleading commonsense knowledge", it is possible that the mistakes are due to difficulty in recognizing negations, etc. For other instruction-tuned larger models, have you tried to ask the model to explain the reason for its answers? Do they have a good understanding of what is a brainteaser?

**Reasons To Accept:**

The proposed dataset is challenging and of good quality according to human validation, which could inspire further studies. The authors construct two adversarial versions considering the models may have seen the data, and their experiments and analyses are detailed.

**Reasons To Reject:**

A. For a challenging task such as this, larger model is preferred over smaller models under zero-shot settings. While the results show that common sense knowledge has limited benefits (or even negative impact), using the RoBERTa model for commonsense-related analysis is not convincing enough to me.

B. While the number of few-shot examples has no clear impact on sentence puzzles, it is possible that the given examples are very different from test examples, i.e., it is possible due to the "not good enough" examples that the model can hardly learn generalizable patterns.

**Reproducibility:**

5: Could easily reproduce the results.

**Reviewer Confidence:**

3: Pretty sure, but there's a chance I missed something. Although I have a good feel for this area in general, I did not carefully check the paper's details, e.g., the math, experimental design, or novelty.

---

> ### Author Rebuttal · Authors · 2023-08-29
>
> We genuinely appreciate your praise of our work and even more the comments and insightful suggestions! Here we list the answers in relation to your comments:
>
> ### Use of RoBERTa Model for Commonsense-Related Analysis:
> We agree that large language models are better choices in zero-shot settings, which is why we include GPT3.5, T0 and FlanT5. As a complementary baseline with clearer training data than LLMs, we study RoBERTa to focus on the impact of common sense in lateral thinking. We select RoBERTa because of its widespread usage of commonsense tasks and its impressive performance on zero-shot evaluations. RoBERTa-L (CSKG) achieve SOTA zero-shot result on multiple commonsense tasks, while CAR shares similar structures with RoBERTa-L (CSKG) and even outperforms ChatGPT on commonsense tasks.
> ### Few-Shot Examples and Generalizable Patterns:
> The goal of the few-shot setting is to introduce the model to the task at hand (lateral thinking in our case) by example. While few-shot examples can provide examples similar on the surface (Line 577-582), these are not helpful for model generalization on this subtask (Line 572-577) because brain teasers in the sentence puzzles are similar on a more abstract level. Retrieving examples based on high-level similarity patterns for overall few-shot prompting performance is non-trivial and it serves as an exciting research direction highlighted by our paper.
> ### Possibility for more nuanced question types (Q.A):
> As shown in Appendix A.1, the reasoning paths and topics within each brain teaser are too diverse to support a reasonable sample of each category. For further insight, please refer to one of the public resource websites (https://www.riddles.com/posts).
> ### Exploration Regarding Commonsense Knowledge and Model Explanations (Q.B):
> Our analysis of the model explanations revealed that the model recognized an estimated 28% of the questions as brain teasers (36% for Word Puzzle, 20% for Sentence Puzzle), based on the model mentions of brain teasers (e.g., “This is a popular brain teaser...”. To study the remaining causes in a more systematic manner, we annotate 5% of the error cases with their reasons and find that 73% of the cases are led by commonsense associations. In 33% of cases, the commonsense distractors lead models to overwrite or ignore the original context in question, while in 40% of cases, models make the wrong choices and make wild imagination due to the stereotype (e.g. Denying sources of illumination implies darkness). While this analysis is not exhaustive, it provides insight into the most common reasons for the model behaviour, revealing that commonsense associations are a frequent cause, which is expected given the training objectives of LLMs. It is certainly possible that the model's associations neglect other aspects of language like negation. We will extend this evaluation with a thorough user study in follow-up work to understand the model behaviour on brain teasers through its explanations.

---

### Official Review · Reviewer_aqde · 2023-08-07

**Soundness:** 4

**Excitement:**

4: Strong: This paper deepens the understanding of some phenomenon or lowers the barriers to an existing research direction.

**Paper Topic And Main Contributions:**

Vertical Thinking tasks involve a sequential analytical process that is based on rationality and logic, whereas a Lateral Thinking task requires a creative thinking process that involves applying a new perspective to the problem statement and defying preconceptions. For instance, the answer to *What type of cheese is made backwards?* is `Edam` which is `made` written backwards.

This paper proposes to study the ability of the state-of-the-art LLMs to reason on lateral thinking puzzles by formulating such puzzles as MCQA tasks, making them intuitive to answer by humans and easy to evaluate automatically. It further describes a novel **BrainTeaser** benchmark involving two tasks with varying granularities: Sentence Puzzles, and Word Puzzles. Authors design a data collection procedure with semi-automatic filtering. To address memorization and inconsistency, the benchmark is enriched with semantic reconstruction and context reconstruction. Results reveal a huge gap b/w human performance and current LLMs.

**Questions For The Authors:**

see weaknesses.

**Reasons To Accept:**

The paper is well written, easy to follow, and fun to read. The experiments are well motivated and laid out clearly.

The paper identifies lateral thinking as a useful/relevant task to push the creative-reasoning abilities of LLMs when the key focus in this general direction has been relatively limited to Vertical Thinking puzzles. The dataset could definitely be of interest to the folks working on commonsense reasoning. The data collection procedure seems sound and the evaluation is adequate.

The results reveal a huge gap b/w human vs. LLM performance, indicating the great opportunity/need to improve lateral thinking in LLMs.

**Reasons To Reject:**

In the process of reviewing this paper, I went through a number of *lateral thinking* puzzles on the internet. While this is indeed an intriguing task, solving these puzzles is a hard task. The answers to the questions are often not very obvious. Consider the following example that I found [online, [1] ](https://web.archive.org/web/20230807191758/https://parade.com/1288259/marynliles/lateral-thinking-puzzles/):

*Three kids enter a room, but only two walkout. The room is empty. Where is the third kid?* --> The answer is: The third kid uses a wheelchair, so they roll out instead of walking out.

As evident, the solution goes well beyond the commonsense aspect of a statement or a human's/machine's *general* reasoning ability. This makes me rethink if the task is reasonable.

To further quench my curiosity, I asked 5 computer science PhD students to answer a few of the examples in the paper and none of them came up with the same answers as described in the paper. (Now, this changed positively after providing choices when they could see the "path" to the right answer more clearly. I also ACK that this not the most scientifically grounded technique to verify an intuition.) This makes me wonder how smart the human/LLM needs to be in order to answer these questions? When getting humans to answer these questions, how were they chosen? If the *average* human is not able to answer these questions, is it reasonable to assume that an LM *should* be able to answer these?

I find these questions compelling when looking at Table 4 with the huge performance gap in the human vs. LLM reasoning.

Finally, while using the MCQA format reduces the difficulty of the task, it increases the chances of spurious correlations or wrong reasons. Table 5 has some good examples.

(While this is not my primary area of research, I have done my best to read the literature and spend time reading the paper. I'm open to be convinced that these are not major problems and that the positives of the work outweigh the negatives.)

[1] https://web.archive.org/web/20230807191758/https://parade.com/1288259/marynliles/lateral-thinking-puzzles/

**Reproducibility:**

4: Could mostly reproduce the results, but there may be some variation because of sample variance or minor variations in their interpretation of the protocol or method.

**Reviewer Confidence:**

2: Willing to defend my evaluation, but it is fairly likely that I missed some details, didn't understand some central points, or can't be sure about the novelty of the work.

---

> ### Author Rebuttal · Authors · 2023-08-29
>
> Thank you for the constructive feedback and for investing the time to explore the content and context of our paper! Here is a list of answer together with a quick pointer to the place in the paper:
>
> ### Reasonableness of the Task:
> We have tried an open-ended format in our initial experiment (Line 218-221) and both human and LLMs perform poorly. Moreover, as the reviews point out (and as we note in Lines 221-223), the puzzles often have multiple creative solutions. Therefore, we consider the MCQA format for a fair comparison and construct a multi-step pipeline to ensure that the task is reliable and the distractors are incorrect (Line 223-227). Similar to the performance improvement observed in your experiment, our main result shows a huge gap between humans and LLMs: the distractors won’t cause confusion to humans, while LLMs perform much worse.  Human participants in our study are chosen from diverse backgrounds to ensure a broad representation. We will provide details on the demographics and selection criteria of the human participants in the appendix.
>
> ### MCQA Format and Spurious Correlations:
> Based on our comparison with an open-ended task format (Line 218-221), the MCQA format has proven to be the most reliable option for presenting the BrainTeaser task. MCQA formats are commonly used for reasoning tasks [1,2,3], including the most similar task to ours, RiddleSense [4]. Our well-motivated definition of brain teasers (Sec. 3.1) and distractor sampling strategy (Sec 3.3) ensure there exists only one reasonable answer, preventing spurious correlations or wrong reasons to our best effort. While we acknowledge that spurious correlations may remain in the data, none of our baseline methods was able to solve the task, which indicates that simple surface patterns are not sufficient.
>
>
> [1] [CommonsenseQA: A Question Answering Challenge Targeting Commonsense Knowledge](https://aclanthology.org/N19-1421) (Talmor et al., NAACL 2019)
>
> [2] [Think you have Solved Question Answering? Try ARC, the AI2 Reasoning Challenge](https://arxiv.org/abs/1803.05457) (Clark et al., arxiv 2018)
>
> [3] [Can a Suit of Armor Conduct Electricity? A New Dataset for Open Book Question Answering](https://aclanthology.org/D18-1260) (Mihaylov et al., EMNLP 2018)
>
> [4] [RiddleSense: Reasoning about Riddle Questions Featuring Linguistic Creativity and Commonsense Knowledge.](https://aclanthology.org/2021.findings-acl.131.pdf) (Lin et al.,  ACL-IJCNLP 2021)

---

### Meta-Review · Area_Chair_4rvh · 2023-09-17

**Recommendation:** 5

**Metareview:**

This work describes and evaluates a novel and interesting benchmark for lateral thinking composed of 1,100 puzzles accompanied by high-quality annotations.

All three reviewers agree that the soundness of the proposal is either good or strong and there is a clear consensus on the excitement being strong for all three too. To the extent possible, I would like to see the reviewers questions & concerns incorporated into the final version of the paper in order to improve its clarity. These include: further explanations on the difficulty of the task itself and the evaluation metrics chosen.

---

### Decision · Program_Chairs · 2023-10-07

**Decision:**

Accept-Main

**Comment:**

This work describes and evaluates a novel and interesting benchmark for lateral thinking composed of 1,100 puzzles accompanied by high-quality annotations.

All three reviewers agree that the soundness of the proposal is either good or strong and there is a clear consensus on the excitement being strong for all three too. To the extent possible, I would like to see the reviewers questions & concerns incorporated into the final version of the paper in order to improve its clarity. These include: further explanations on the difficulty of the task itself and the evaluation metrics chosen.